# Renal Cell Carcinoma Surgical Treatment Disparities in American Indian/Alaska Natives and Hispanic Americans in Arizona

**DOI:** 10.3390/ijerph19031185

**Published:** 2022-01-21

**Authors:** Francine C. Gachupin, Benjamin R. Lee, Juan Chipollini, Kathryn R. Pulling, Alejandro Cruz, Ava C. Wong, Celina I. Valencia, Chiu-Hsieh Hsu, Ken Batai

**Affiliations:** 1Department of Family and Community Medicine, College of Medicine, University of Arizona, Tucson, AZ 85711, USA; celina@arizona.edu; 2Department of Urology, College of Medicine, University of Arizona, Tucson, AZ 85724, USA; brlee@arizona.edu (B.R.L.); Jchipollini@arizona.edu (J.C.); kpulling@email.arizona.edu (K.R.P.); acruz700@surgery.arizona.edu (A.C.); Awong@email.arizona.edu (A.C.W.); kbatai@email.arizona.edu (K.B.); 3Department of Epidemiology and Biostatistics, Mel and Enid Zuckerman College of Public Health, University of Arizona, Tucson, AZ 85724, USA; pchhsu@arizona.edu

**Keywords:** kidney cancer, cancer health disparities, nephrectomy, surgical treatment, Arizona cancer

## Abstract

American Indians/Alaska Natives (AI/AN) and Hispanic Americans (HA) have higher kidney cancer incidence and mortality rates compared to non-Hispanic Whites (NHW). Herein, we describe the disparity in renal cell carcinoma (RCC) surgical treatment for AI/AN and HA and the potential association with mortality in Arizona. A total of 5111 stage I RCC cases diagnosed between 2007 and 2016 from the Arizona Cancer Registry were included. Statistical analyses were performed to test the association of race/ethnicity with surgical treatment pattern and overall mortality, adjusting for patients’ demographic, healthcare access, and socioeconomic factors. AI/AN were diagnosed 6 years younger than NHW and were more likely to receive radical rather than partial nephrectomy (OR 1.49 95% CI: 1.07–2.07) compared to NHW. Mexican Americans had increased odds of not undergoing surgical treatment (OR 1.66, 95% CI: 1.08–2.53). Analysis showed that not undergoing surgical treatment and undergoing radical nephrectomy were statistically significantly associated with higher overall mortality (HR 1.82 95% CI: 1.21–2.76 and HR 1.59 95% CI: 1.30–1.95 respectively). Mexican Americans, particularly U.S.-born Mexican Americans, had an increased risk for overall mortality and RCC-specific mortality even after adjusting for neighborhood socioeconomic factors and surgical treatment patterns. Although statistically not significant after adjusting for neighborhood-level socioeconomic factors and surgical treatment patterns, AI/AN had an elevated risk of mortality.

## 1. Introduction

Across the United States (U.S.), American Indians/Alaska Natives (AI/AN) and Hispanic Americans (HA) have higher kidney cancer incidence and mortality rates compared to non-Hispanic Whites (NHW) [1,2,3]. Additionally, in Arizona, AI/AN and HA experience a significantly higher incidence of kidney cancer compared to NHW, with AI/AN men having kidney cancer as the second most commonly diagnosed cancer with a 1.9-fold higher incidence [4]. Clear-cell renal cell carcinoma (RCC) has been reported to be more common in AI/AN [5] than NHW, and the mean age at diagnosis among AI/AN was 49.7 years, with a 6-fold higher odds of diagnosis at a younger age [6]. HA also have higher incidence and mortality in Arizona. Analysis of RCC suggests that sociodemographic factors, such as location of residence (urban/rural) and neighborhood factors, including Census-track high school graduation, unemployment, and poverty rates [7], have influenced cancer diagnosis in Arizona. However, these two racial/ethnic minority groups are underrepresented in RCC clinical studies [8]. The two most common surgical treatment approaches in the U.S. are partial and radical nephrectomy for localized kidney cancer. The experience for treatment among AI/AN and HA is not well understood.

In this article, we describe the disparity in RCC surgical treatment for AI/AN and HA and its potential association with mortality in Arizona. We also examined whether neighborhood-level socioeconomic factors account for the disparities in surgical treatment and mortality.

## 2. Materials and Methods

RCC case data were obtained from the Arizona Cancer Registry (ACR). The ACR is a population-based registry which includes cancer cases reported from state-licensed hospitals. This study focused on stage I RCC cases diagnosed between 2007 and 2016. Partial nephrectomy that conserves renal function is currently recommended for patients with cT1a renal mass to minimize the risk of chronic kidney disease, while radical nephrectomy is used for the removal of a larger tumor and more complex tumors. Local ablation is an alternative approach for patients with a small renal mass who may not be a good candidate to undergo nephrectomy [9]. Due to heterogeneity, other kidney cancer subtypes were excluded. This study included only AI/AN, NHW, HA, non-Hispanic Blacks (NHB), and Asian Americans. Cases with unknown or other race/ethnicity were excluded. The ACR case data were linked to the U.S. 2000 Census-tract socioeconomic measures, including high school graduation, poverty, and unemployment rates and the 2021 Rural-Urban Continuum Codes (RUCC). For analysis of de-identified data, the University of Arizona IRB considered our study exempt (#1710934077).

Descriptive statistics were done for stage I RCC patient characteristics based on race/ethnicity of the patients for age, sex, marital status, education, employment, poverty rate, and RUCC. Descriptive statistics for RCC grade, histologic subtype, and recurrence were also completed. Because the sample size for NHB and Asian Americans was small, they were excluded from subsequent statistical analysis. Logistic regression analyses were done to analyze associations between race/ethnicity with no surgical treatment using NHW as the reference, first adjusting for age, sex, marital status, RCC subtype, and year of diagnosis (Model 1). Then, neighborhood characteristics (high school graduation, poverty, and unemployment rates and RUCC) were added to the regression model (Model 2). Logistic regression analyses were also done to test associations between race/ethnicity and radical nephrectomy, adjusting for age, sex, marital status, RCC subtype and diagnosis year in Model 1 and adding adjustments of education, unemployment, and poverty in Model 2. Logistic regression analysis was performed to assess the association between race/ethnicity and recurrence because the date of recurrence was not obtained for the ACR. Cox regression analyses were performed to assess associations between race/ethnicity and overall mortality, adjusted for age, sex, marital status, RCC subtype, grade, and diagnosis year in Model 1; education, poverty, as well as no surgical treatment vs. treatment or radical vs. partial nephrectomy were added to Model 2 and Model 3. Cox regression analysis was also performed to examine the effect of nephrectomy type on risk of mortality by stratifying patients’ nephrectomy type. Sub-distribution Cox proportional hazards regression was performed to evaluate time to death due to RCC, accounting for competing risks.

## 3. Results

### 3.1. Patient Sociodemographic Characteristics

A total of 5111 patients from the ACR were included in the analyses: 319 (6.2%) AI/AN, 3728 (72.9%) NHW, 168 (3.3%) NHB, 37 (0.7%) Asian Americans, and 859 (16.8%) HA (Table 1). The largest racial/ethnic group in the study were NHWs, followed by HA and AI/AN, respectively. Both AI/AN and HA were diagnosed at a median age of 59 years, 6 years younger than NHW. More males than females (61.9% vs. 39.1%) were diagnosed across all racial/ethnic groups, with 59.2% male patients among AI/AN, compared to 62.0% for NHW and 56.2% for HA. Although most RCC patients tended to be married, the percent of patients reporting being married was lowest among AI/ANs (41.7% vs. 64.4% for NHW and 55.8% for HA). The percent reporting being single (30.1%) or of unknown marital status (10.0%) were highest among AI/AN. AI/AN (11.3%) and HA (13.5%) were less likely to live in neighborhoods with higher high school graduation rates (vs. 48.6% for Asian Americans and 40.5% for NHW). The percent living in neighborhoods with high unemployment rates was highest among AI/AN (54.7% vs. 9.6% for NHB and 18.6% for HA). Similarly, the percent living in neighborhoods with high poverty rates was highest among AI/AN (69.2% vs. 39.7% for HA and 9.0% for NHW). A higher proportion of AI/AN patients live in rural areas than patients from other racial/ethnic groups (51.4% vs. 21.3% for HA and 18.9% for NHW). There were statistically significant differences across all sociodemographic variables by race/ethnicity. Among HA, Mexican Americans had very different sociodemographic characteristics from NHW (Appendix A). Compared to U.S.-born Mexican Americans, Mexico-born Mexican Americans were more likely to live in neighborhoods with low socioeconomic characteristics.

### 3.2. RCC Patient Characteristics

Most AI/AN were diagnosed with Grade 1 or 2 RCC (83.2% vs. 74.7% for NHW and 76.4% for HA) (Table 2). Clear cell RCC was the most common histologic subtype across most racial/ethnic groups, including AI/AN (89.9%) and HA (86.1%). However, NHB (39.5%) had more papillary histologic subtypes (41.2%), which was the second most common histologic subtype for the other racial/ethnic groups. There were statistically significant differences for RCC histologic subtypes across the race/ethnicity groups. The recurrence rate was highest among AI/AN (19.4% vs. 18.2% for Asian Americans and 14.3% for NHW). Overall, recurrence was less common in HA (13.9%) compared to NHW, but recurrence occurred at higher frequency among individuals of Mexican descent (21.0%; *p* = 0.008), particularly in U.S.-born Mexican Americans (22.7%). 

### 3.3. Associations between Race/Ethnicity and No Surgical Treatment

Across all race/ethnic groups, AI/AN were more likely to have no treatment (13.9% in AI/AN vs. 7.7% in NHW and 8.3% in HA; Table 3). Nephrectomy was the most common treatment, with AI/AN receiving this treatment at the lowest rates (72.2% vs. 82% for NHW and 83% for HA). In logistic regression models assessing the association between race/ethnicity and not undergoing treatment (no treatment vs. local ablation or nephrectomy), controlling for age, sex, marital status, RCC subtype and diagnosis year, AI/AN patients were statistically significantly more likely to receive no treatment (OR 2.18, 95% CI 1.50–3.16). When neighborhood characteristics were added (Model 2), the association was no longer significant (*p* = 0.31). HAs were further sub-categorized to Mexican Americans, and not undergoing treatment was found to be statistically significant (OR 1.66, 95% CI: 1.08–2.53). This finding continued even after the model distinguished between U.S.-born and Mexico-born Mexican Americans and with the addition of neighborhood-level high school graduation and unemployment rates.

### 3.4. Associations between Race/Ethnicity and Radical Nephrectomy

Radical nephrectomy was more common among AI/AN (59.1%) and HA (50.4%) compared to NHW (48.3%; Table 4). In logistic regression models assessing the association between race/ethnicity and nephrectomy type (radical vs. partial), controlling for age, sex, marital status, RCC subtype and diagnosis year, AI/AN patients were statistically significantly more likely to receive radical nephrectomy (OR 1.69, 95% CI 1.25–2.27). Adding education, unemployment and poverty in the model attenuated the association, but the association remained statistically significant (OR 1.49, 95% CI 1.07–2.07).

### 3.5. Recurrence

AI/AN did not have increased odds of recurrence, but there were increased odds of recurrence in individuals of Mexican descent (Appendix A). The odds of recurrence in Mexican Americans compared to NHW were greater among patients that underwent radical nephrectomy (OR 3.12, 95% CI 1.00–9.76) than patients who underwent partial nephrectomy (OR 1.31, 95% CI 0.46–3.71; Appendix A). 

### 3.6. Cox Regression Analysis for Mortality in the Arizona Cancer Registry

Cox regression analyses were conducted to assess whether surgical treatment and nephrectomy type, controlling for age, sex, marital status, RCC subtype, grade, and diagnosis year, has a bearing on overall mortality, and the results were statistically significant for AI/AN patients and patients of Mexican descent, whether born in the U.S. or Mexico (Table 5). Although it was not statistically significant for AI/AN in fully adjusted model, increased risk of overall mortality persisted for AI/AN and U.S.-born Mexican Americans when surgical treatment, education and poverty were added to the model. Mexican Americans, particularly U.S.-born Mexican Americans, had significantly elevated risk of RCC-specific mortality, adjusting for competing risk, surgical treatment, and socioeconomic factors, while the association was not significant for AI/AN (Appendix A). In a stratified analysis based on nephrectomy type, a similar pattern of increased risk of overall mortality in Mexican Americans was observed in patients who underwent radical nephrectomy (HR 2.41, 95% CI: 1.56–3.72) and partial nephrectomy (HR 2.37, 95% CI: 1.37–4.09; Appendix A).

## 4. Discussion

The goal of this study was to describe the disparity in surgical treatment for AI/AN and HA diagnosed with RCC and to assess any potential association with mortality. The results demonstrate that AI/AN and HA patients are more likely to be younger and to live in neighborhoods with low high school graduation rates and high poverty and unemployment rates. Compared to U.S.-born Mexican Americans, Mexico-born Mexican Americans were more likely to live in neighborhoods with low socioeconomic characteristics, which may be partially due to ongoing immigration policy and legislation. These socioeconomic factors likely impact housing, cancer care support and healthcare access [10,11]. Furthermore, married individuals are known to have better health outcomes, and AI/AN had relatively high unmarried single status, which may partially explain the reported treatment results [12,13].

Recurrence was not statistically significant for AI/AN. Although HA recurrence was less common compared to NHW, recurrence occurred at a higher frequency among U.S.-born Mexican Americans [14,15]. Both AI/AN and HA were more likely to receive no treatment compared to NHW; however, neighborhood characteristics likely explained the findings for AI/AN but only partially for Mexican Americans. For those that did receive treatment, AI/AN tended to receive a more invasive surgical treatment, radical nephrectomy. Neighborhood-level socioeconomic factors may partly account for the disparities in surgical treatment for AI/AN. Many AI/AN utilize the Indian Health Service (IHS) or the local tribal health services program, and surgical treatment disparities may be attributed to payor status. If an AI/AN patient was referred for surgical treatment by either the IHS or the local tribal health services program, the payment may have dictated the type of reimbursable care. The personal experiences of RCC diagnosis and surgical treatment among AI/AN RCC patients were not part of our current study and would be important to include in future work. Nevertheless, the standard of care for small renal masses is partial nephrectomy; hence, results could be attributed to local referral patterns and lack of access to academic hospitals or healthcare systems with comprehensive cancer care, where minimally invasive surgical treatment and more advanced oncology care are available [16].

The survival benefit of undergoing partial nephrectomy instead of the more invasive radical nephrectomy is well demonstrated [17]. For patients with multiple comorbidities that impact kidney health, partial nephrectomy that preserves kidney function would be optimal [18,19]. However, HA, especially U.S.-born Mexican Americans, did have statistically increased risk of overall mortality after including surgical treatment patterns and neighborhood-level socioeconomic factors. The reasons for high recurrence and mortality rates in U.S.-born Mexican Americans and increased odds of recurrence among Mexican Americans who underwent radical nephrectomy are not well understood. Acculturation to Western lifestyle may have partially influenced cancer progression [15,17], but future studies need to investigate the prevalence of comorbidities and obesity in this population and the potential impact on RCC progression. Improving social determinants of health and providing equitable surgical treatment may reduce disparities in mortality for HA. Our study did not assess access to and use of local cancer navigator services or translation or interpretation services and the potential impact of these services on advocacy. 

Our findings underscore that disease characteristics, socioeconomic factors and treatment differences do not completely explain the disparities in care and treatment that AI/AN experience. This suggests there is a need for more detailed information in the kidney cancer databases of other risk factors, including smoking, obesity, diabetes, hypertension, and chronic kidney disease, which are known risk factors of RCC pathogenesis and potential factors that influence survival after surgical treatment [20]. We know from other cancers [21,22] and other chronic conditions [23,24] that these risks are disproportionately higher among AI/AN [25]. We also know that the cancer experiences for AI/AN individuals and families are challenging [26] and that comprehensive programs to address AI/AN cancer health disparities are needed [27,28,29]. It is also important to promote a healthy lifestyle, and for those individuals with a family history of kidney cancer, it becomes important to schedule annual exams, especially for men. 

This study also demonstrated surgical treatment disparities in HA, but surgical treatment disparities may not be the primary factor for high mortality rates in individuals of Mexican descent. Mexican Americans consistently had an elevated risk of mortality across different analyses in this study. Similar to AI/AN, obesity and comorbidities are also common in Mexican Americans and may have influenced survival after the surgical treatment [30]. Mexican Americans also share similar clinicopathologic characteristics with AI/AN, including early age of diagnosis and predominantly clear cell type of RCC [5], as well as varying degrees of Indigenous American ancestry [31]. HA may have an underlying genetic and biologic basis of RCC pathogenesis and progression. However, they may be Mexican American-specific risk factors, and future studies are necessary to assess the factors contributing to surgical treatment disparities and higher mortality in Mexican Americans.

Several limitations of the current study include the lack of detailed clinical information in ACR data, for example, complexity of tumor, comorbidities, performance status (measurement of patient’s daily activities without the help of others), and smoking history, that may be important for clinical decision-making for surgical treatment and have mortality implications. Individual-level socioeconomic or healthcare access factors were not used in this study, and these factors may additionally account for treatment disparities identified in this study. While AI/AN are the third largest racial/ethnic group in Arizona, the number of AI/AN patients included in this study was small. This may have led to false-negative associations in the fully adjusted regression models. Furthermore, the ACR has information on whether patients were treated or referred from IHS facilities, but there may be potential misclassification of additional AI/AN cases [32]. Missing cases could have underestimated the magnitude of surgical treatment disparities, rate of recurrence, and mortality [33].

Future directions could include an analysis of the hereditary patterns and molecular analysis of the surgical specimen to determine if there are genetic predispositions and biologic characteristics of tumor progression related to disparate oncologic outcomes in AI/AN. However, this would be challenging due to continuing concerns by Tribes of the use of biological samples in research projects based on past transgressions [34,35,36,37]. To ensure AI/AN populations and individuals diagnosed with cancer fully benefit from medical advances and to be comfortable to participate in research, respectful adherence to Tribal Sovereignty needs to be an integral facet of collaborative partnership building [38,39,40].

## 5. Conclusions

AI/AN and HA were diagnosed 6 years younger; they were more likely to receive no treatment, or if treatment was received, it was more likely to be radical nephrectomy, which was associated with higher overall mortality. It seems for both AI/AN and HA, structural inequality is also an RCC risk factor. AI/AN and HA were more likely to live in neighborhoods with lower high school graduation, higher unemployment, and higher poverty rates and rural areas, thereby underscoring that social determinants of health continue to play a key role in cancer health disparities. In this study, we were able to provide information on some key AI/AN and HA RCC disparities, but more research is needed. 

Studies that further our understanding of potential points of intervention should involve patient experiences, surgeon decision-making, and healthcare system treatment processing, with the end goal being the provision of equitable RCC care, especially since the challenges that AI/AN and HA RCC patients face when they try to get surgical treatment have never been discussed anywhere in the literature.

## Figures and Tables

**Table 1 ijerph-19-01185-t001:** Stage I RCC patient characteristics across racial/ethnic groups in the Arizona Cancer Registry.

Stage 1 RCC Patient Characteristics	AI/AN	NHW	NHB	Asian American	HA	*p*
	(*n* = 319)	(*n* = 3728)	(*n* = 168)	(*n* = 37)	(*n* = 859)	
Age, Median (IQR)	59 (49–67)	65 (55–72)	58 (50–66)	61 (48–72)	59 (48–67)	<0.001
Sex, *n* (%)						0.006
Male	189 (59.2)	2311 (62.0)	113 (67.3)	26 (70.3)	483 (56.2)	
Female	130 (40.8)	1417 (38.0)	55 (32.7)	11 (29.7)	376 (43.8)	
Marital Status, *n* (%)						<0.001
Married	133 (41.7)	2400 (64.4)	95 (56.5)	26 (70.3)	479 (55.8)	
Single	96 (30.1)	506 (13.6)	42 (25.0)	5 (13.5)	164 (19.1)	
Separated/Divorced/Widowed	58 (18.2)	738 (19.8)	27 (16.1)	5 (13.5)	190 (22.1)	
Unknown	32 (10.0)	84 (2.3)	4 (2.4)	1 (2.7)	26 (3.0)	
High School Education, *n* (%)						<0.001
≥90%	36 (11.3)	1507 (40.5)	54 (32.3)	18 (48.6)	116 (13.5)	
≥70%, <90%	69 (21.7)	1872 (50.3)	79 (47.3)	13 (35.1)	355 (41.3)	
<70%	213 (67.0)	346 (9.3)	34 (20.4)	6 (16.2)	388 (45.2)	
Unemployment, *n* (%)						<0.001
<5%	71 (22.3)	2313 (62.1)	89 (53.3)	23 (62.2)	292 (34.0)	
≥5, <10%	73 (23.0)	1273 (34.2)	62 (37.1)	14 (37.8)	407 (47.4)	
≥10%	174 (54.7)	139 (3.7)	16 (9.6)	0 (0.0)	160 (18.6)	
Poverty Rate, *n* (%)						<0.001
<25%	51 (16.0)	2294 (61.6)	76 (45.5)	23 (62.2)	244 (28.4)	
≥25, <50%	47 (14.8)	1097 (29.4)	51 (30.5)	11 (29.7)	274 (31.9)	
≥50%	220 (69.2)	334 (9.0)	40 (24.0)	3 (8.1)	341 (39.7)	
RUCC 2013, *n* (%)						<0.001
1 or 2	155 (48.6)	3022 (81.1)	157 (94.0)	34 (91.9)	676 (78.7)	
3–7	164 (51.4)	704 (18.9)	10 (6.0)	3 (8.1)	183 (21.3)	

**Table 2 ijerph-19-01185-t002:** Stage I RCC characteristics across racial/ethnic groups in Arizona Cancer Registry.

Stage 1 RCC Characteristics	AI/AN	NHW	NHB	Asian Americans	HA	*p*
	(*n* = 319)	(*n* = 3728)	(*n* = 168)	(*n* = 37)	(*n* = 859)	
Grade, *n* (%)						0.03
1 & 2	168 (83.2)	2107 (74.7)	96 (82.1)	20 (71.4)	505 (76.4)	
3 & 4	34 (16.8)	713 (25.3)	21 (17.9)	8 (28.6)	156 (23.6)	
RCC Histologic Subtype, *n* (%)						<0.001
Clear Cell	204 (89.9)	2099 (73.0)	47 (39.5)	21 (70.0)	552 (86.1)	
Papillary	12 (5.3)	435 (15.1)	49 (41.2)	6 (20.0)	39 (6.1)	
Chromophobe	4 (1.8)	232 (8.1)	14 (11.8)	2 (6.7)	29 (4.5)	
Others	7 (3.1)	108 (3.8)	9 (7.6)	1 (3.3)	21 (3.3)	
Recurrence, *n* (%)						0.24
No	200 (80.6)	2650 (85.7)	123 (84.6)	27 (84.8)	614 (86.1)	
Yes	48 (19.4)	441 (14.3)	22 (15.3)	6 (18.2)	99 (13.9)	

**Table 3 ijerph-19-01185-t003:** Associations between race/ethnicity and no surgical treatment.

				Model 1		Model 2	
Race/Ethnicity	No Treatment, *n* (%)	Local Abrasion, *n* (%)	Nephrectomy, *n* (%)	OR (95% CI)	*p*	OR (95% CI)	*p*
NHW vs. racial/ethnic minority groups							
NHW	286 (7.7)	380 (10.2)	3048 (82.1)	Reference		Reference	
AI/AN	44 (13.9)	44 (13.9)	229 (72.2)	2.18 (1.50–3.16)	<0.001	1.26 (0.80–1.99)	0.31
HA	71 (8.3)	75 (8.8)	710 (82.9)	1.35 (1.01–1.80)	0.04	1.02 (0.74–1.41)	0.90
NHW vs. Mexican Americans							
NHW				Reference		Reference	
Mexican Americans	40 (14.1)	22 (7.8)	221 (78.1)	2.21 (1.51–3.25)	<0.001	1.66 (1.08–2.53)	0.02
NHW vs. U.S./Mexico-born Mexican Americans							
NHW				Reference		Reference	
U.S.-born Mexican Americans	23 (19.2)	15 (12.5)	82 (68.3)	2.61 (1.55–4.37)	<0.001	2.00 (1.17–3.34)	0.01
Mexico-born Mexican Americans	8 (8.4)	6 (6.3)	81 (85.3)	1.42 (0.66–3.03)	0.37	0.95 (0.43–2.13)	0.91

Model 1 includes age category, sex, marital status, and diagnosis year; Model 2 includes age category, sex, marital status, diagnosis year, RUCC 2013, % high school graduation, % unemployment, and % poverty.

**Table 4 ijerph-19-01185-t004:** Associations between race/ethnicity and radical nephrectomy.

			Model 1		Model 2	
Race/Ethnicity	Radical, *n* (%)	Partial, *n* (%)	OR (95% CI)	*p*	OR (95% CI)	*p*
NHW vs. racial/ethnic minority groups						
NHW	1296 (48.3)	1389 (51.7)	Reference		Reference	
AI/AN	123 (59.1)	85 (40.9)	1.69 (1.25–2.27)	0.001	1.49 (1.07–2.07)	0.02
HA	314 (50.4)	309 (49.6)	1.14 (0.95–1.36)	0.15	1.09 (0.90–1.32)	0.36
NHW vs. Mexican Americans						
NHW			Reference			
Mexican Americans	100 (50.3)	99 (49.7)	1.05 (0.78–1.41)	0.75	0.95 (0.70–1.30)	0.77
NHW vs. U.S./Mexico-born Mexican Americans						
NHW			Reference		Reference	
U.S.-born Mexican Americans	41 (56.2)	32 (43.8)	1.26 (0.78–2.04)	0.34	1.18 (0.72–1.91)	0.52
Mexico-born Mexican Americans	33 (46.5)	38 (53.5)	0.88 (0.54–1.42)	0.59	0.78 (0.47–1.28)	0.33

Model 1 includes age category, sex, marital status, RCC subtypes (including NOS) and diagnosis year; Model 2 includes age category, sex, marital status, RCC subtypes (including NOS) and diagnosis year, % high school graduation, % unemployment, and % poverty.

**Table 5 ijerph-19-01185-t005:** Cox regression analysis for overall mortality in ACR.

Variables	Adjusted Model 1		Adjusted Model 2		Adjusted Model 3	
	HR (95% CI)	*p*	HR (95% CI)	*p*	HR (95% CI)	*p*
Surgical Treatment						
Local Ablation or Nephrectomy			Reference			
No Treatment			1.82 (1.21–2.76)	0.04		
Nephrectomy Type						
Partial					Reference	
Radical					1.59 (1.30–1.95)	<0.001
Race/ethnicity						
NHW	Reference		Reference		Reference	
AI/AN	1.82 (1.29–2.55)	0.001	1.42 (0.98–2.06)	0.06	1.52 (0.96–2.39)	0.07
Hispanic Americans	1.27 (1.01–1.62)	0.04	1.05 (0.81–1.36)	0.70	1.27 (0.95–1.71)	0.11
NHW vs. Mexican Americans						
NHW	Reference		Reference		Reference	
Mexican Americans	2.65 (2.06–3.39)	<0.001	2.28 (1.72–3.04)	<0.001	2.53 (1.81–3.52)	<0.001
NHW vs. U.S./Mexico-Born Mexican Americans						
NHW	Reference		Reference		Reference	
U.S.-Born Mexican Americans	4.01 (3.00–5.36)	<0.001	3.54 (2.56–4.89)	<0.001	4.08 (2.80–5.95)	<0.001
Mexico-Born Mexican Americans	1.73 (1.05–2.86)	0.03	1.51 (0.90–2.56)	0.12	1.67 (0.93–3.02)	0.09

Model 1: Adjusted for age category, sex, marital status, RCC subtypes (excluding NOS), grade (1 and 2 vs. 3 and 4), diagnosis year (categorical); Model 2: Model 1 + % high school graduation., % poverty, and no surgical treatment vs. treatment; Model 3: Model 2 + % high school graduation., % poverty, and radical vs. partial nephrectomy.

## Data Availability

The RCC case data used in the current study can be requested from the ACR.

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
