# Peer review of "Renal Cell Carcinoma Surgical Treatment Disparities in American Indian/Alaska Natives and Hispanic Americans in Arizona"

_ijerph, 2022, doi:10.3390/ijerph19031185_

Round 1
Reviewer 1 Report
- This is a interesting study, however authors do not present well and comprehensively.
- In the introduction line 55 ." This paper has a special focus on AI/AN who.. " Why not focus on HA ( US-born Mexican American) ??
- All study need IRB approval. De-identified data still need informed consent waiver approval.
- Te married status influences the study results. AN/AI with relative high unmarried and single status . Need to explain in discussion .
- In results line 109 .-110 : Similarly, the percent living in neighborhoods with high poverty rate was highest 109 among AI/AN (69.2% vs. 39.7% for HA and 90% for NHW" Are you sure 90 % for NHW ??
- In results line 144: please present statistical reference for significance.
- Table 3 . NHWt is typo
- Table 5. you should explain why RN has higher mortality than PN in the discussion .
- In discussion line 218: " AI/AN did not have statistically increased risk of overall mortality after including surgical treatment patterns and neighborhood-level socioeconomic factors." But HA, Mexican American, especially US born Mexican American do . You should address these issue.
- The title is disparity in AI/AN and HA in surgical treatment, but HA result and finding do not demonstrate in conclusion .
- Authors sub-classify Mexican American to Mexico born Mexican American and US- born Mexican Americans. and find the difference. Why not sub-classify and separate to AI and AN respectively.
- Discussion need to explain why radical nephrectomy has high recurrence rate than partial nephrecotmy.. Why US born Mexican American has poor results than Mexico-born Mexican American.
- Typo need to be corrected. Discussion is weak .
Author Response
1. This is a interesting study, however authors do not present well and comprehensively.
Thank you for your time and input to improve our manuscript to be more comprehensive and to be presented better.
2. In the introduction line 55 ." This paper has a special focus on AI/AN who.. " Why not focus on HA ( US-born Mexican American) ??
The reference to having AI/AN be a special focus group has been deleted throughout the paper.
3. All study need IRB approval. De-identified data still need informed consent waiver approval.
Our study was considered exempt by the University of Arizona IRB and the sentence has been revised to reflect this review more accurately.
4. Te married status influences the study results. AN/AI with relative high unmarried and single status . Need to explain in discussion .
You are absolutely correct and a reference that married individuals have higher compliance with health care access and treatment regimens and the results for AI/AN with high single status may partially explain the treatment results we found, has been added.
5. In results line 109 .-110 : Similarly, the percent living in neighborhoods with high poverty rate was highest 109 among AI/AN (69.2% vs. 39.7% for HA and 90% for NHW" Are you sure 90 % for NHW ??
Thank you. The percent for NHW was missing the decimal point and has been changed to 9.0%.
6. In results line 144: please present statistical reference for significance.
The statistical reference has been added.
7. Table 3 . NHWt is typo
Thank you. The typo has been corrected.
8. Table 5. you should explain why RN has higher mortality than PN in the discussion .
We added discussion on survival benefit of partial nephrectomy, particular to preserve renal function for patients with multiple comorbidities.
9. In discussion line 218: " AI/AN did not have statistically increased risk of overall mortality after including surgical treatment patterns and neighborhood-level socioeconomic factors." But HA, Mexican American, especially US born Mexican American do . You should address these issue.
The discussion has been edited to reflect the surgical treatment patterns and neighborhood-level socioeconomic factors for HA.
10. The title is disparity in AI/AN and HA in surgical treatment, but HA result and finding do not demonstrate in conclusion .
The conclusion has been edited to reflect that surgical treatment for HA is not comparable to NHW.
11. Authors sub-classify Mexican American to Mexico born Mexican American and US- born Mexican Americans. and find the difference. Why not sub-classify and separate to AI and AN respectively.
Thank you for the suggestion; however, the AI/AN race classification is set by the US Bureau of the Census and, unless primary data collection is initiated with the distinction, we have no choice but to use the existing categorization.
12. Discussion need to explain why radical nephrectomy has high recurrence rate than partial nephrecotmy.. Why US born Mexican American has poor results than Mexico-born Mexican American.
The reasons for these observed disparities in Mexican Americans are unknown and are still under-investigation. We added this to this discussion section.
13. Typo need to be corrected. Discussion is weak .
Edits have been made to the discussion with the intention to make our findings stronger. Typos have been corrected.
Reviewer 2 Report
The authors present a well written manuscript that is clear and easy to read. One issue requires clarification: it is not clear if the IRB approved the study as exempt or how data were linked to external databases. Was an honest broker used since zip code likely was required and is a HIPAA protected variable?
Author Response
1. The authors present a well written manuscript that is clear and easy to read. One issue requires clarification: it is not clear if the IRB approved the study as exempt or how data were linked to external databases. Was an honest broker used since zip code likely was required and is a HIPAA protected variable?
Thank you for your review.
Our study was considered exempt by the University of Arizona IRB and the sentence has been revised to reflect this review more accurately. Our study did not include an honest broker. You are correct that zip code is a HIPAA identifier.
Reviewer 3 Report
I read with interest the manuscript from Gachupin and coworkers, focusing on disparities in treatment of renal cancer between different racial/ethnic groups in Arizona. Globally, it is a well-written paper, with a proper statistical analysis and a balanced discussion.
Some comments as follows:
- Abstract: In the study, data of AI, AN, and AH (American Hispanic) were compared to those of NHW (as also specified in title). However, no results regarding AH were reported. I invite the authors to include in abstract some outcomes relate to that subgroup.
- Introduction: Similar to the previous comment. In the last sentence “This paper has a special focus on AI/AN who have highest RCC mortality in the U.S. but are underrepresented in RCC studies”, however the authors stated that also incidence and mortality in the HA is higher in Arizona. The authors should be better defining the relevance of HA ethnic group in their work, considering that HA represented the second largest racial/ethnical group (16.8%) of the study population.
- Table 2 and paragraph: R.2 the authors stated that “Recurrence rate was highest among AI/AN ... Overall, recurrence was less common in HA (13.9%) compared to NHW, but recurrence occurred at high frequency among individuals of Mexican descent (21.0%), particularly in U.S.-born Mexican Americans (22.7%).” However, according to table 2, the recurrence rate is not significantly different between groups. This should be underscored.
- Table 3. There are a couple of errors to be corrected (NHWt in the first column; ‘local abrasion’). Four-hundred and ninety nine patients received local ablation. I wonder whether their tumours were included or not in the histology analysis, since histologic diagnosis in renal cancer is usually made after surgical removal of the specimen, and preoperative percutaneous core biopsy is not mandatory.
- See comments written in abstract and introduction about the HA subgroup. The last sentence ‘Studies such as…’ is a little bit long and can be rephrased.
Author Response
1. Abstract: In the study, data of AI, AN, and AH (American Hispanic) were compared to those of NHW (as also specified in title). However, no results regarding AH were reported. I invite the authors to include in abstract some outcomes relate to that subgroup.
Thank you for this important comment. The reference to having AI/AN be a special focus group has been deleted throughout the paper. The findings specific to HA have been better defined, including American Hispanics.
2. Introduction: Similar to the previous comment. In the last sentence “This paper has a special focus on AI/AN who have highest RCC mortality in the U.S. but are underrepresented in RCC studies”, however the authors stated that also incidence and mortality in the HA is higher in Arizona. The authors should be better defining the relevance of HA ethnic group in their work, considering that HA represented the second largest racial/ethnical group (16.8%) of the study population.
Thank you for this important comment. The reference to having AI/AN be a special focus group has been deleted throughout the paper. The findings specific to HA have been better defined.
3. Table 2 and paragraph: R.2 the authors stated that “Recurrence rate was highest among AI/AN ... Overall, recurrence was less common in HA (13.9%) compared to NHW, but recurrence occurred at high frequency among individuals of Mexican descent (21.0%), particularly in U.S.-born Mexican Americans (22.7%).” However, according to table 2, the recurrence rate is not significantly different between groups. This should be underscored.
Recurrence rate was statistically significantly higher in Mexican American compared to NHW (p=0.008, Supplementary Table 1). We added p-value to the text.
4. Table 3. There are a couple of errors to be corrected (NHWt in the first column; ‘local abrasion’). Four-hundred and ninety nine patients received local ablation. I wonder whether their tumours were included or not in the histology analysis, since histologic diagnosis in renal cancer is usually made after surgical removal of the specimen, and preoperative percutaneous core biopsy is not mandatory.
Thank you. The typo has been corrected. Thank you for this comment. In fact, histology information is missing for many patients who received local abrasion. RCC subtype probably did not affect clinical decision making for whether patients should undergo surgical treatment or not. We reanalyzed removing RCC subtype in the regression model. In this current version, we present analysis results of regression model removing RCC subtype.
5. See comments written in abstract and introduction about the HA subgroup. The last sentence ‘Studies such as…’ is a little bit long and can be rephrased.
Thank you. The sentence has been rephrased.
Round 2
Reviewer 1 Report
I still insist the necessity of IRB approval .
Author claim " Our study is considered exempt by the University of Arizona IRB ...."
As I stated previously, all studies need IRB approval and some studies could be informed consent waiver. Please offer IRB approval number .
Author Response
I still insist the necessity of IRB approval .
Author claim " Our study is considered exempt by the University of Arizona IRB ...."
As I stated previously, all studies need IRB approval and some studies could be informed consent waiver. Please offer IRB approval number .
Thank you for your human subjects protection advocacy. You are correct that our study was submitted and reviewed by the University of Arizona IRB. The assigned protocol number is 1710934077. The IRB determined that IRB review was not required.